# Attention based automated radiology report generation using CNN and LSTM

**Mehreen Sirshar**[1], **Muhammad Faheem Khalil Paracha**[1]*, **Muhammad Usman Akram**[1],
**Norah Saleh Alghamdi**[2], **Syeda Zainab Yousuf Zaidi**[3], **Tatheer Fatima**[4]

**1** Department of Computer and Software Engineering, College of Electrical and Mechanical, National University of Sciences and Technology, Islamabad, Pakistan, **2** College of Computer and Information Sciences, Princess Nourah Bint Abdulrahman University, Riyadh, Saudi Arabia, **3** Department of Computer Science, Bahria University, Islamabad, Pakistan, **4** Resident Radiologist, Pakistan Institute of Medical Sciences, Islamabad, Pakistan

* faheem.paracha18@ce.ceme.edu.pk

**Data Availability Statement:** Both datasets are publicly available for researchers and can be accessed using following links: a. https://www.kaggle.com/raddar/chest-xrays-indiana-university b. https://openi.nlm.nih.gov/detailedresult?img=

## Abstract

The automated generation of radiology reports provides X-rays and has tremendous potential to enhance the clinical diagnosis of diseases in patients. A new research direction is gaining increasing attention that involves the use of hybrid approaches based on natural language processing and computer vision techniques to create auto medical report generation systems. The auto report generator, producing radiology reports, will significantly reduce the burden on doctors and assist them in writing manual reports. Because the sensitivity of chest X-ray (CXR) findings provided by existing techniques not adequately accurate, producing comprehensive explanations for medical photographs remains a difficult task. A novel approach to address this issue was proposed, based on the continuous integration of convolutional neural networks and long short-term memory for detecting diseases, followed by the attention mechanism for sequence generation based on these diseases. Experimental results obtained by using the Indiana University CXR and MIMIC-CXR datasets showed that the proposed model attained the current state-of-the-art efficiency as opposed to other solutions of the baseline. BLEU-1, BLEU-2, BLEU-3, and BLEU-4 were used as the evaluation metrics.

## 1. Introduction

Chest diseases are fatal to human life. Common chest diseases such as pneumonia, pneumothorax, and effusion [1] are diagnosed with the help of medical images, such as chest X-rays (CXR) and CT scans. These images provide subsequent evidence of chest abnormalities captured through a proper pathological process. A radiologist conducts an analytical examination for the presence of even a minor abnormality on an X-ray image, followed by a detailed diagnostic textual report of a patient. This manually created report (see Fig 1) describes the condition of the chest in general, detailed findings, and diseases, if they are projected on the X-ray image. Writing medical reports is a laborious task. In developing countries with a large population with poor health conditions, such as Pakistan, radiologists may have to capture

CXR111_IM-0076-1001&req=4 c. https://physionet.org/content/mimic-cxr/2.0.0/. For the Mimic dataset, researchers have to complete some initial formalities to justify their research potential and then they can download the data. There are no other concerns associated with these datasets.

**Funding:** This research project was funded by the Deanship of Scientific Research, Princess Nourah bint Abdulrahman University, through the "Program of Research Project Funding After Publication, grant No (42-PRFA-P-53)". https://www.pnu.edu.sa/en/Pages/home.aspx. The funders had no role in study design, data collection and analysis, decision to publish, or preparation of the manuscript.

**Competing interests:** The authors have declared that no competing interests exist.

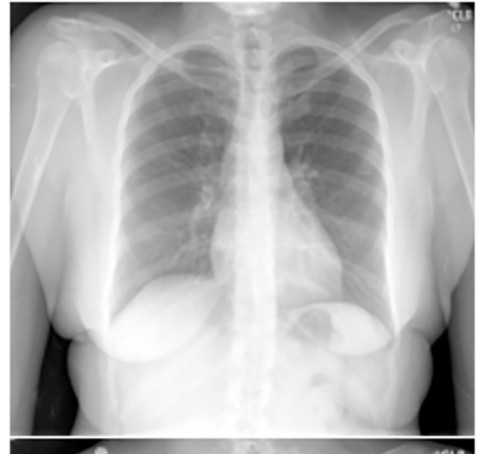

**Doctor Findings:**
The lungs are clear bilaterally.
No evidence of focal consolidation, pneumothorax, or pleural effusion.

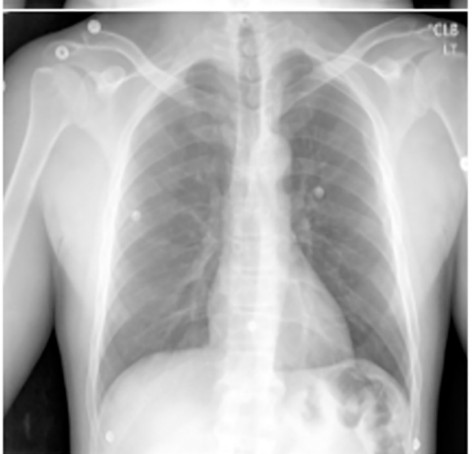

**Doctor Findings:**
Left-sided rib deformities consistent with old fractures.
Mild degenerative changes about the thoracolumbar junction.

**Fig 1. Examples of chest X-ray images and radiology reports.**

hundreds of X-ray images of different patients every day. Generating hundreds of reports on pathological conditions of lungs against CXR is time-consuming and tedious. The process of describing X-rays in terms of text is not efficient, even for specialist doctors in their respective fields. Moreover, this task is error-prone due to inexperienced radiologists, faulty reasoning by radiologists, staff shortage in hospitals, or additional workload in the hospitals that cause errors in the reports [2].

Additionally, writing accurate reports is very difficult task for the pathologists and radiologists with less experience and for those working in rural areas with barely any healthcare facilities. To properly read and understand a radiograph, the following skills are needed [3]. (i) Knowledge about the basic physiology of chest diseases and other information about any normality or abnormality of thorax anatomy; (ii) the ability to find the association with other indicative diseases (respiratory function tests, test results, and electrocardiograms); (iii) the ability to understand the changes in the radiographs over time; (iv) familiarity with patient clinical history; and (v) the ability to analyze radiographs through a fixed pattern. In other words, writing medical reports is a strenuous task for both experienced and inexperienced medical professionals. The proposed research is thus derived from the motivation to improve the clinical diagnostic systems by adding the functionality to generate reports automatically. The current automatic report generation approaches suffer from various limitations that need to be addressed to complete this task.

The first limitation is the understanding of diseases that appear as white projections of some well-understood patterns on the CXR, and then applying language semantics to express these in natural languages such as English. Therefore, in addition to visual understanding, a natural language processing model is required for report generation.

In contrast to the existing models, the proposed research presents a model to solve the problems of visual representation as well as sentence generation. As a first step, the proposed model takes CXR images as input $I$ and performs the feature extraction process for disease identification. In the second step, the model is trained to generate the desired report, which consists of LSTM followed by an attention mechanism. To optimize the report, the probability $p(S|I)$ is determined, where $S = \{S_1, S_2, S_3. . .\}$ represents a set of words generated for the report from the vocabulary that sufficiently defines the contents in the CXR images [4].

The proposed model is motivated by the recent advancements in machine translation, where the goal is to transform the source composed of a sequence of tokens to the targeted sequence of tokens by maximizing the likelihood $p(T|S)$, where $S$ is the sequence of tokens present in a source space and T is the targeted sequence of words.

The remainder of this paper is organized as follows. Section 2 provides a review of the literature and significant work done by researchers in the past few years. Section 3 describes the proposed methodology in detail. Section 4 presents all the datasets and experimental results in detail with the relevant figures and tables. Finally, Section 5 concludes the paper.

## 2. Related work

In recent years, several chest radiograph datasets have been made publicly available. A summary of all of these datasets is presented in Table 1. A number of researchers have worked on caption generation for general images and detailed report creation for medical images. Tanti et al. [5] classified generative models into two types: (i) injection architecture and (ii) merge architecture. In the injection architecture, the input is the tokenized captions and the image vectors to an RNN block, whereas in the merge architecture, the input is only the captions to the RNN block, and merges the output with the effective image learning computational models by leveraging the information in the medical images and the free-text reports in the emerging field. Such a combination of image and textual data helps to further improve the model performance in automatic report generation (Litjens et al.) [6]. Correctly reading the CXR images is exasperating due to the huge variability, variation, and complexity of the diseases as well as their treatments, using computerized tomography (CT) scans (Rubin, 2015) [7].

Schlegl et al. [13] first proposed a weakly supervised learning approach to utilize semantic descriptions in the reports as labels for better classifying tissue patterns in OCT imaging. They specified how accurate voxel level classifiers would be and how this information increases the classification accuracy for intraretinal SRF, IRC, and normal retinal tissues. In 2015, Shin et al.

**Table 1. Summarized specification of publically available chest X-ray datasets.**

| Dataset | Source Institution | Disease Labeling | No of Images | No of Reports | No of Patients |
|---|---|---|---|---|---|
| IU Chest X-Ray (Demner-Fushman et al. [8]) | Indiana Network for Patient Care | Expert | 8,121 | 3,996 | 3,996 |
| MIMIC-CXR (Johnson et al. [9]) | Beth Israel Deacones Medical Center | Automatic (CheXpert labeler) | 4,73,057 | 2,06,563 | 63,478 |
| Chest-XRay8 (Wang et al. [10]) | National Institutes of Health | Automatic (DNorm + MetaMap) | 1,08,948 | - | 32,717 |
| PadChest (Bustos et al. [11]) | Hospital Universitario de San Juan | Expert + Automatic (Neural network) | 1,60,868 | 2,06,222 | 67,625 |
| CheXpert (Irvin et al. [12]) | Stanford Hospital | Automatic (CheXpert labeler) | 2,24,316 | - | 65,240 |

[14] and Wang et al. [10, 15] proposed a network that comprises a CNN and RNN in the field of radiology that is jointly trained to find abnormalities in CXR. They mined the radiological reports to create disease and symptom concepts as labels. They first used LDA to find the topics for clustering, and then applied disease detection tools such as DNorm, MetaMap, and several other NLP tools for downstream CXR classification using a convolutional neural network. They also released a label set along with image data. Later, Wang et al. [16] used the same exact CXR dataset to further improve the performance of disease classification and report generation from medical images.

For report generation, Jing et al. [17] built a multi-task learning framework, which consists of co-attention and a hierarchical LSTM that predicts the tags, localizes the regions with abnormalities, and uses these for the radiological image annotation and report paragraph generation. They performed their experiments on two publicly available datasets: IU CXR [8] and PEIR Gross [17]. Moradi et al. [18] jointly processed image and text signals to produce CXR images of regions of interest. They proposed two architectures to find their region of interest in CXR and then to generate a textual report. One of these architectures is comprised of CNN and LSTM, and its training was performed using images, their corresponding reports, and the markings of regions of interest (ROIs) for those X-rays; the second one consists of a pre-trained network on a large dataset of the same type of images for feature learning to obtain their findings of interest. Rubin et al. [7] trained a convolutional network to predict common thoracic diseases using CXR images. They proposed a novel architecture called DuelNet that processes both frontal and lateral X-ray images while emulating routine clinical practice. The dataset used was the MIMIC dataset, which is almost four times larger than the size of the largest previously used CXR dataset (ChestX-Ray8) [10].

Li et al. [19] suggested a reinforcement learning-based named HRGR agent to train the report generator to decide whether to make a sentence using a template or generate a new sentence. This work was believed to be the first to combine human prior knowledge and generative neural networks at the same time to generate medical reports. This agent was updated using reinforcement learning. Alternatively, Gale et al. [20] generated interpretable hip fracture X-ray reports by identifying image features and filling text templates. It comprises the training of a simple RNN model to produce hip fracture reports to clarify the results of the neural network classifiers.

Finally, Hsu et al. [21] proposed a model in which he trained radiological images and reported joint representation through unsupervised alignment of the cross-modal embedding spaces via both local and global information retrieval. Experiments were performed on the MIMIC dataset, which contains both medical images and their corresponding reports.

Machine translation has already been performed for several years by defining a sequence of different activities, such as independently translating terms, aligning phrases, and reordering; however, recent developments have suggested easier and better ways to perform the same tasks by utilizing a recurrent neural network (RNN) [22–24], which provides state-of-the-art performance. RNN is composed of two parts: encoder and decoder. The encoder reads the source sequences that may be either text or images and then transforms them into a vector representation of a fixed length, which then acts as the initial hidden state of the decoder that produces the targeted sequence of words.

The proposed model applies a deep convolutional neural network (CNN) as an encoder to an RNN. This encoder converts the input CXR into a vector representation of a fixed length for use in multiple computer vision tasks [5]. The CNN encoder obtains the details about CXR contents that are used as the input to the decoder LSTM followed by the attention block, which efficiently generates the medical reports (see Fig 3).

The main contributions of the proposed research in the medical report generation are

- An innovative model that provides end-to-end solutions for the problem with state-of-the-art sub-networks, CNN as an encoder, and LSTM followed by attention as a decoder.

- An entirely trainable neural network utilizing vision features along with attention heads for better report generation

- Finally, substantial experiments on the IU and MIMIC CXR dataset demonstrating the significance of our proposed approach.

## 3. Model

A probabilistic and neural-network-based model is proposed to produce the radiograph report. Recent advancements in the computational machine translation have demonstrated that with a strong sequence model, the state-of-the-art outcomes can be obtained by explicitly optimizing the probability of the successful translation in an end to end manner, provided as an input sequence, both for the training and the inference. Such models use an RNN that converts a variable-size input of the encoder into a fixed size vector. The fixed-size representation is then used as an input to the decoder part to convert this into a meaningful appropriate sequence of words. Thus, in the proposed model, the variable size input is CXR, the encoder is CNN, and the decoder is LSTM, followed by attention, which uses the same source as the target language conversion principle.

The main objective is to directly maximize the likelihood of accuracy of the medical report, as originally described by a radiologist or pathologist. This is achieved by the mathematical formulation represented in Eq 1.

$$\theta^* = \arg \max_{\theta} \sum_{(I,S)} \log p(S|I; \theta) \tag{1}$$

In the above equation, $\theta^*$ and $\theta$ are considered as the parameters of the proposed model. S is the correct medical report of CXR I. Remember S can be a sentence of any length. Therefore, the chain rule has been considered as one of the easiest approaches to obtain the combined likelihood from $S_0$ to $S_N$, where N is the maximum number of words in the report. This can be represented by Eq 2.

$$\log p(S|I) = \sum_{t=0}^{N} \log p(S_t|I, S_0, S_1, S2 \ldots S_{t-1}) \tag{2}$$

For ease, the dependency would be on $\theta$. Training pairs are created for training $(S, I)$. The goal is to maximize the sum of the log probabilities of S over T over all the training pairs and to optimize this using the gradient descent, as described in (2). Further details regarding the training are discussed in section 4.

It is natural to implement $p(S_0, S_1 \ldots S_{t-1})$ with LSTM, where different numbers of words in a sentence (up to t-1) as described in Eq 2 are stated with the help of a hidden state of a fixed length or a memory unit $h_t$. By using the nonlinear function f after obtaining a new input $x_t$, the hidden state or memory is updated. This is stated in Eq 3 as

$$h_{t+1} = f(h_t, x_t) \tag{3}$$

Two important structural decisions must be made to render the LSTM more workable. First, what type of functions would be appropriate for the model, as well as how it can manage both the input CXR images and words to the same system. To provide the structural decision, the suggested model uses a specific type of network called the long short-term memory

(LSTM) network. LSTM has already been proven as the best network when sequence-related tasks, such as translation, must be performed. However, before feeding the input of the encoder to LSTM, the input is passed to the attention mechanism, whose main purpose is to focus only on those parts of the images that are of our region of interest and have maximum information.

A CNN was applied to describe the contents of the CXR images. CNN has already proved itself as the current state-of-the-art network for visual classification or image-related tasks, and the VGG16 architecture of CNN is selected because it is based on a novel approach of batch normalization and won the ILSVRC 2014 classification competition [25]. In addition, it generalized many tasks using transfer learning, such as scene classification [26]. The words are used in the system with the help of an embedded model through which they are converted into vectors using one hot scheme.

## 3.1 Convolutional Neural Network (CNN)

CNN is a special type of neural network that provides sophisticated performance in image processing and visual representation tasks. Some of the best applications of CNN are feature extraction and classification based on those features, such as image segmentation, object detection, etc. The CNN is composed of different types of convolutional layers. Similar to the multi-layer neural network [27], there are fully connected (FC) layers after these convolutional layers. A CNN is built in such a way as to take advantage of the 2D input image structure. With the support of multiple local ties and linked weights, this task is accomplished along with many pooling methods that translate the input data into invariant features. The key benefit of CNN includes the freedom to prepare and offer fewer parameters than other networks with the same number of hidden states.

The visual geometry group (VGG) network, which is a deep convolution neural network for large-scale visual recognition, was used in this study [28]. The VGG has many variants. The most famous are VGG16 and VGG19, which have 16 and 19 layers, respectively. The classification errors for both VGG16 and VGG19 were almost the same for both the validation data and the test data, which were 7.4% and 7.3%, respectively.

The proposed model used the transfer learning approach to train the VGG16 architecture, as shown in Fig 2, to efficiently extract the features from the input images (CXR) using a

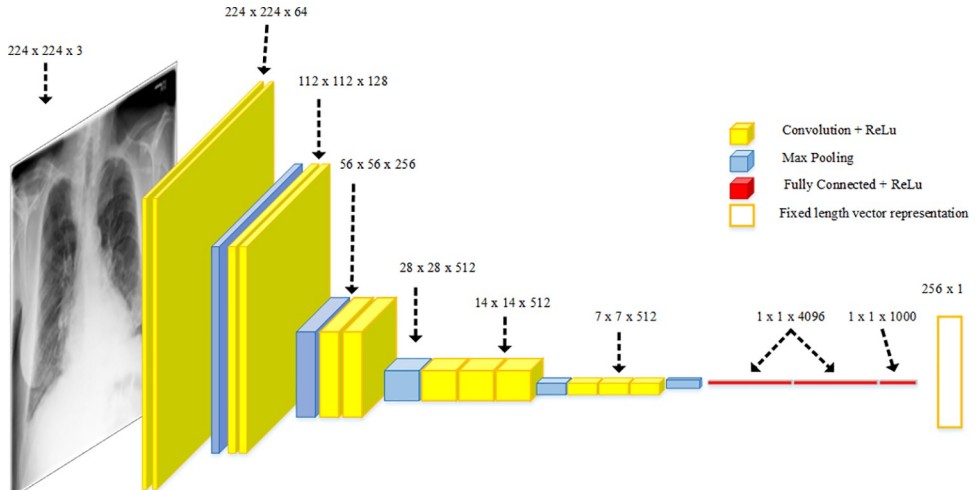

**Fig 2. VGG-16 neural network architecture with highlighted sizes and each layer units.**

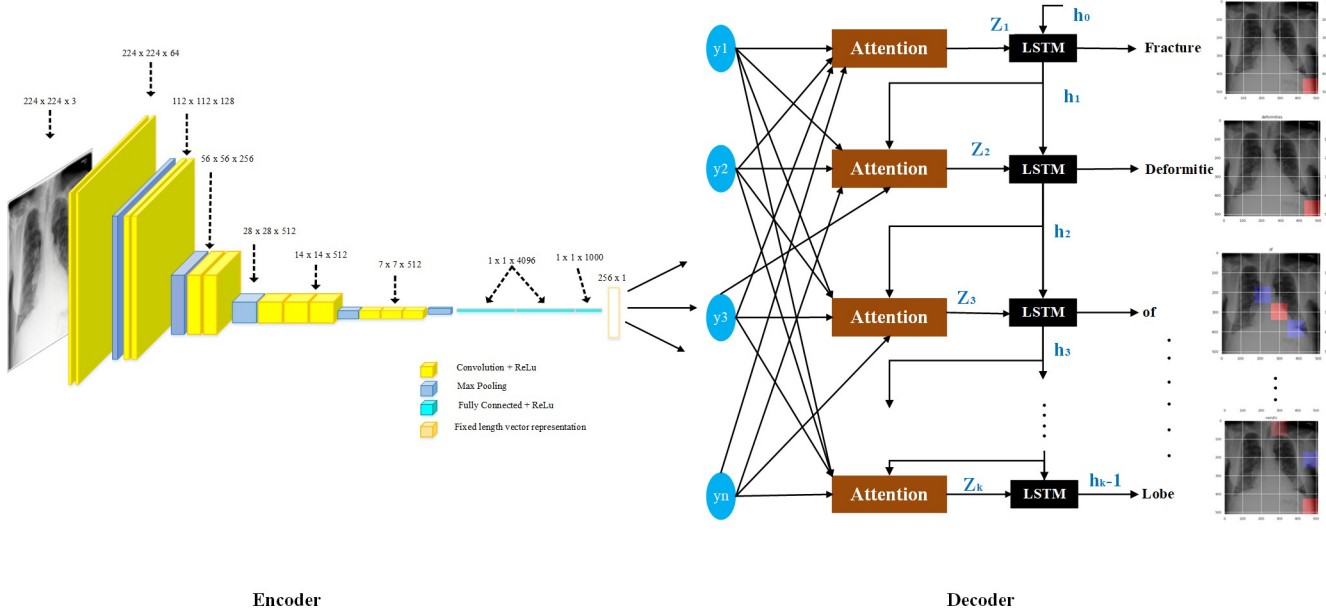

**Fig 3. VGG-16 architecture and associated parameters.**

combination of multiple $3 \times 3$ convolution layers and max pooling layers. The Softmax layer of VGG16 was replaced with the final $1 \times 4096$ FC layer. This layer now acts as an input to the decoder as well as the generation of medical reports. The output of the VGG16 network is a vector of size $1 \times 4096$, which will later be converted into a fixed vector length of $1 \times 256$ that is used to represent the features of the images.

A dropout layer was added to the network with a value of 0.5, to reduce overfitting. An optimal value is between 0.5 and 0.8, which indicates the probability at which the outputs of the layer are dropped out. A dense layer is added after the dropout layer, which basically applies the activation function to the input, the kernel with a bias. The activation function used was rectified linear units (ReLU), and the size of the output space was specified as 256. These vectors of size 256 are the output of the feature extraction model, which will then be used as the input of the attention block followed by LSTM. Fig 3 shows detailed architecture of VGG-16 along with all parameters.

## 3.2 Word embedding

Word embedding is primarily responsible for processing the captions of each image given as input during the training process. The output of the word embedding is also a vector of size $1 \times 256$, which is another input to the decoder sequences.

Initially, the captions present along with each CXR were tokenized. Tokenization is a process through which the words in these sentences are converted to integers so that the neural network can process them efficiently. The tokenized captions are padded to ensure that the length of all sentences is equal to the size of the longest sentence with max words.

Then, an embedding layer is attached to embed the tokenized captions into fixed dense vectors with an output space of $256 \times 22$. 22 was chosen as the maximum number of words in all the findings of the IU CXR dataset. These vectors further ease the processing by providing a convenient way to represent words in the vector space. A dropout layer is attached again with a probability of 0.5, to reduce overfitting in the model.

### 3.3 Attention mechanism

The attention mechanism helps the LSTM to focus on just part of an image, which is a specific and of interest, when generating a new word while completing a caption. Thus, it is simply saying that a decoder is going to generate the caption by using only some specific information.

Consequently, with this new block of attention in the architecture, we are going to predict the next word of the medical report by not only the hidden state of the decoder, but also using the context vector that contains information of our interest. Therefore, we divide the image into n parts; then, at the i-th location of the report, we use the $h_i$ hidden state of LSTM. So now, this $h_i$ is used as the context to select the relevant part of the radiograph. The attention model output is denoted by $z_i$. The output $z_i$ can be considered as a vector that contains only those parts of the image that have the main information and our point of interest, and it is now easy for LSTM to generate a new word that actually describes the content and the diseases present in CXR and the relationship between them. One most important thing to consider is that after LSTM generates a new word, it also returns a new hidden state $h_{t+1}$ for the generation of the next word, and so on. The mathematical representation of the attention mechanism is as follows.

$$e_{jt} = f_{aTT}(s_{t-1}, h_j) \tag{4}$$

- In the equation, $e_{jt}$ means at every i-th timestamp of the decoder and the importance of the j-th pixel location in the input image.

- $s_{t-1}$ is the previous state of decoder

- $h_j$ is the state of encoder

$f_{aTT}$ is a simple feed forward a neural network which is a linear transformation of input $U_{attn}{}^* h_j + W_{attn}{}^* s_t$ and then, a non-linearity (tanh) on top of that, an again one more transformation $V_{attn}^T$. This is a scalar quantity.

$$f_{aTT} = V_{attn}^T * tanh(U_{attn} * h_j + W_{attn} * s_t) \tag{5}$$

Now, when we know the input, we need to feed the weighted sum combination of input to the decoder.

$$C_t = \sum_{j=1}^{T} \alpha_{jt} * h_j \ such \ that \ \sum_{j=1}^{T_x} \alpha_{jt} = 1 \tag{6}$$

$$where \ \ \alpha_{ij} \geq 0$$

where $C_t$ is the context vector.

$$s_t = RNN(s_{t-1}, [e(\hat{y}_{t-1}), c_t]) \tag{7}$$

Here,

- $s_{t-1}$ is the previous state of the decoder

- $e(\hat{y}_{t-1})$ is the previous predicted word

- $C_t$ is the context vector, i.e., the weighted sum of the input.

Finally, we can say that it is a better modeling technique than the previous one, and it is said to be a more informed model because we are trying to obtain our results in a more natural way.

## 3.4 Long Short-Term Memory (LSTM)

It is difficult for a simple RNN to develop long-term stability when the problem of vanishing gradients and exploding gradients is very common [29]. To overcome this problem, a specific type of recurrent network called LSTM was introduced [29] and successfully extended for translation tasks [23, 30] and sequence generation [31].

**3.4.1 LSTM based sentence generator.** The main idea behind the LSTM model is memory cell c, which encodes information based on what input is observed at any time (see Fig 4). The operation of the cell is controlled by "gates" or layers that are inserted multiplicatively and can retain either values coming from the gates as 0 or 1. In particular, three gates are being used to track if the current value of the cell should be forgotten, whether the new cell value (output gate o) is to be produced, or to be interpreted as its input. Eqs 8, 9 and 10 represent the input, forget, and output layers, respectively, where Eqs 11, 12 and 13 represent the other operations of LSTM [29].

$$i_t = \sigma(W_{ix}x_t + W_{im}m_{t-1}) \tag{8}$$

$$f_t = \sigma(W_{fx}x_t + W_{fm}m_{t-1}) \tag{9}$$

$$o_t = \sigma(W_{ox}x_t + W_{om}m_{t-1}) \tag{10}$$

$$c_t = f_t \odot c_{t-1} + i_t \odot h(W_{cx}x_t + W_{cm}m_{t-1}) \tag{11}$$

| Input Image | | | Layer | Out | Param |
|---|---|---|---|---|---|
| 224 | 224 | 3 | conv3-64 | 64 | 1792 |
| 224 | 224 | 64 | conv3064 | 64 | 36928 |
| 224 | 224 | 64 | maxpool | 64 | 0 |
| 112 | 112 | 64 | conv3-128 | 128 | 73856 |
| 112 | 112 | 128 | conv3-128 | 128 | 147584 |
| 112 | 112 | 128 | maxpool | 128 | 65664 |
| 56 | 56 | 128 | conv3-256 | 256 | 295168 |
| 56 | 56 | 256 | conv3-256 | 256 | 590080 |
| 56 | 56 | 256 | conv3-256 | 256 | 590080 |
| 56 | 56 | 256 | maxpool | 256 | 0 |
| 28 | 28 | 256 | conv3-512 | 512 | 1180160 |
| 28 | 28 | 512 | conv3-512 | 512 | 2359808 |
| 28 | 28 | 512 | conv3-512 | 512 | 2359808 |
| 28 | 28 | 512 | maxpool | 512 | 0 |
| 14 | 14 | 512 | conv3-512 | 512 | 2359808 |
| 14 | 14 | 512 | conv3-512 | 512 | 2359808 |
| 14 | 14 | 512 | conv3-512 | 512 | 2359808 |
| 14 | 14 | 512 | maxpool | 512 | 0 |
| 1 | 1 | 25088 | fc | 4096 | 102764544 |
| 1 | 1 | 4096 | fc | 4096 | 16781312 |
| 1 | 1 | 4096 | fc | 1000 | 4097000 |
| 1 | 1 | 1000 | Output | 256 | 256256 |
| **Total** | | | | | **138,679,464** |

**Fig 4. Complete and combined model of CNN image embedder and LSTM with the word embedding.** The LSTM is shown in the unrolled version. All LSTMs are using the same parameters.

$$c_t = o_t \odot c_t \tag{12}$$

$$p_{t+1} = Softmax(m_t) \tag{13}$$

The weight (W) metric represents the trained parameters and $\odot$ represents the multiplication with gate value.

These multiplicative gates enable the LSTM to be robustly trained as these gates cope well with the gradients, including burst and vanish [29].

The nonlinearities are hyperbolic tangent h($\cdot$) and sigmoid $\sigma$ ($\cdot$). The $m_t$ used in the last equation is fed to the Softmax function, whose primary purpose is to generate a distribution of a likelihood $p_t$ over all the words present in the vocabulary.

### 3.5 Training

LSTM is trained to guess each word of the report after seeing CXR and all corresponding words, as described by $p(S_t|I, S_0, S_1...S_{t-1})$. To gain more accuracy, it is better to place many copies of the LSTM. A replica of the LSTM was generated for the image. For each term where all LSTM modules share the same parameters, the word is predicted by LSTM at a time t again serving as an input to the attention block, and then the output of that attention block is used as the input to LSTM at time t+1, and so on (see Fig 5). This is instructive for this reason. In the unrolled version, all recurrent connections are converted into feed-forward links. If the input CXR image is denoted as I, and S = $(S_0, S_1...S_N)$ is a correct medical report describing CXR in more depth, the unrolling method can be represented using Eqs 14, 15 and 16, as stated below.

$$x_{-1} = CNN(I) \tag{14}$$

$$x_t = word\ Embedding(We) * S_t\ where$$

$$t \in \{0...N-1\} \tag{15}$$

$$p_{t+1} = LSTM(Attention(x_t)),\ t \in \{0...N-1\} \tag{16}$$

Every word $S_t$ is represented in one hot vector scheme, where the size of the vector for one word will be equal to the size of the vocabulary. Two words, $S_0$ and $S_N$, show the start and end of the medical report. $S_0$ is "startseq" for start and $S_N$ is "endseq" for the end of the medical report. Specifically, the LSTM signal for a full report was produced by emitting the stop term "endseq." Images and words were projected onto the same space. Images are mapped using a CNN, where the words are generated by using the word embeddings. The input CXR image is given only once after passing from the attention block, initially at t = -1, to tell LSTM about the disease present in the radiograph.

It has been experimentally verified that giving the input image at every time step produces poor results as at each phase, the network may have to directly tackle the noise at each time stamp; thus, it is less effective and causes overfitting.

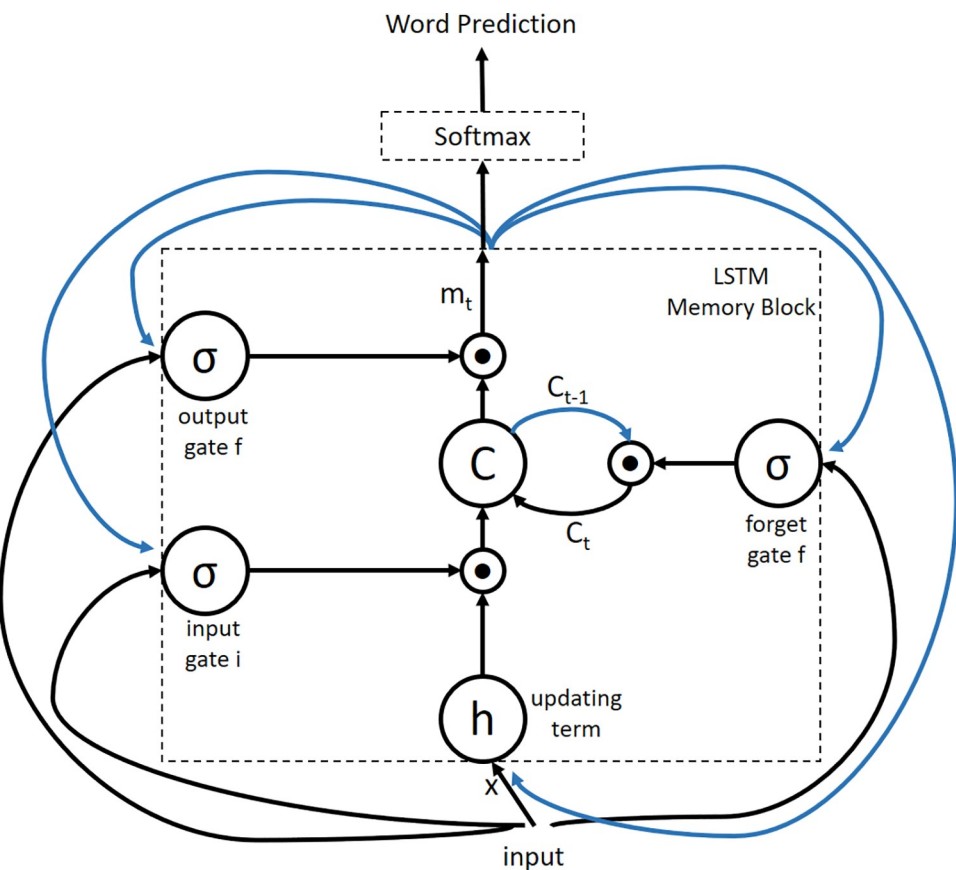

**Fig 5. Long Short-Term Memory (LSTM) network architecture: In the above diagram the memory block comprises a cell c which is essentially controlled by three gates.** These gates are the input, the forget and the output gates.

The loss of the stated model can be calculated by summing the negative log probability of the right term at each time stamp, as described below in Eq 17.

$$L\left(I, S\right) = -\sum_{t=1}^{N} \log p_t(S_t) \tag{17}$$

The loss calculated using the above formula is minimized with regard to many parameters of LSTM, the attention block, CNN, and word embedding. The hyper-parameters and respective configuration are given in Table 2.

**Table 2. Proposed technique hyper parameters and related configuration.**

| Hyper-parameters | Configuration |
|---|---|
| Layers | Encoder (VGG16) + 1 Dense + Decoder-LSTM (9) |
| Optimizer | Adam |
| Activation Function | Relu |
| Learning Rate | 0.001 |
| Batch Size | 64 |
| Loss function | Sparse Categorical Cross entropy |
| Number of attention heads | 1 |
| Dropout rate | 0.5 |

### 3.6 Inference

Many approaches can be used to produce a medical report that provides a radiograph. One of them is Beam Search, in which the selection is performed iteratively by a collection of the best sentences up to time t as candidates to produce size t + 1 sentences and holding only the best k results.

Another approach is sampling in which the 1st word of the report depends on the highest probability $p_1$ in the vocabulary. Then, the embedding of the same previous word is used as the input, and the next word is selected using probability $p_2$. This process is continued until embedding is performed for the end-of-sentence token, where the maximum length depends on the condition. For the experimentation, sampling was used and discussed in the following section.

## 4. Experimentation

We performed a systematic series of studies to test the proposed model's efficacy by comparing the previously developed models as well as with the help of metrics such as the BLEU score.

### 4.1 Evaluation metrics

It has already been discussed that describing CXRs is a difficult task. An experienced radiologist was required to read the CXR. Human evaluation is more efficient than image captioning through natural language. Prior research has proposed many evaluation matrices to check the performance of the proposed model.

Human evaluation is a technique used to measure the performance of a model. However, this is not possible in our case, as we have already discussed the difficulties faced in correctly reading the CXR [2]. The most widely used metric in the research on sentence generation using images has been the BLEU score [32], which is a type of word precision n-gram between produced and referenced reports. For the proposed architecture, we measured the BLEU score to check the accuracy of the proposed model. The possible range of the BLEU score is 0.00 to 1.00. The higher the BLEU score, the better the generation of medical reports because it is basically the comparison of candidate sentences and reference sentences. The candidate sentence is predicted, and the reference sentence is the actual one.

Four types of the BLEU scores were observed: BLEU-1 (1.0, 0, 0, 0), BLEU-2 (0.5, 0.5, 0, 0), BLEU-3 (0.33, 0.33, 0.33, 0) and BLEU-4 (0.25, 0.25, 0.25, 0.25). In addition, cumulative weights have been used because they provide better output. The Adam optimizer [33] was used for parameter learning. Researchers are focusing on this subject and have identified other metrics that are considered more relevant for medical report assessment. We note only one such metric, BLEU, hoping for even further debate and work to come up with a reference to the metric preference.

### 4.2 Dataset

In order to show the validity of proposed technique through detailed experimentations and comparisons, we have used two publicly available CXR datasets i.e. The Indiana University CXR dataset [8] and MIMIC CXR dataset [9]. The Indiana University Chest X-Ray dataset (IU X-Ray) by Demner-Fushman et al. is a collection of CXRs combined with their corresponding medical records. The file format for the X-rays used in this dataset was PNG with a resolution of 512x624 having 24 bits depth. The dataset includes 3,955 radiology reports from two major health networks within the archive of the Indiana Network for Medical Care and 7,470 related CXRs [8]. Almost every report consists of two CXRs of patients, including frontal and lateral views. Different sections in the dataset include the impression, findings, comparison, and

indication. In this research, we used the findings of doctors as the target medical reports to be generated (Fig 1 provides an example).

The second dataset used in this research is MIMIC CXR dataset by Johnson et al. [9]. This is one of the largest publicly available CXR dataset which also contains free-text reports. It includes imaging studies for 65,379 patients, 377,110 CXR images and 227,835 radiology reports. The dataset contains JPEG images of varying sizes having 8bits depth. There are detailed reports along with images in this dataset and we have utilized *findings* section from this whole data as this includes radiology reports against each image.

For both datasets, the first step is to pre-process the data by converting the long findings of the doctors into a short report by converting them into multiple chunks which leads to more than one report associated with each CXR. In addition, all tokens in the reports are converted to lowercase, and all tokens that are not alphabetical are removed. In all experimentations, randomly selected 80% of data is used for training and remaining 20% is used for testing and we repeated the experiment 10 times to show average performance of the proposed technique.

### 4.3 Baselines

The proposed model is compared with different current best performing architectures, for example, LRCN [34] by Donahue et al., Soft ATT [35] by Xu et al., ATT-RK [36] by You et al., and Hieratical Generation [37] by Krause, J et al. Donahue et al. [34] presented a long term RCNN network which consider visual representations along with descriptions. They focused more on RCNN based spatial temporal layers instead of fixed spatio-temporal receptive field. Xu et al. [35] presented a soft attention based model along with visual features for report generation. Their model has the capability to automatically learn to fix its gaze to salient objects while generating reports. A semantic attention based model was introduced by You et al. [36]. Their model learnt to selectively put attention to semantic concept proposals and fused them along with RNN to generate reports. Kause et al. [37] extended same concept presented by You et al. [36] and added hierarchal RNN model for robust captioning. However, subsequent detailed comparative analysis showed that all these baseline methods fell short on analyzing long sentences. We implemented all these models for the radiology report generation and decided to use VGG-16 [28] as the CNN encoder while keeping in mind that these models were built for a short sentence-based report.

### 4.4 Quantitative results

We report the results of the medical report generator results using the standard image captioning evaluation metric, that is, BLEU [32].

We performed some experiments by replacing LSTM with a gate recurring unit (GRU) and bidirectional LSTM by keeping the same VGG16 for the feature extraction and the attention block to work on that part of the image that is of interest. Various images were tested using the above three methods, that is, VGG + LSTM, VGG + GRU, and VGG + Bi-directional LSTM. The training process is accomplished for the above-mentioned models, and the results are obtained and clearly identified that the VGG + LSTM model is more accurate than the other techniques. A comparison of the BLEU scores of all three techniques is presented in Table 3.

In Table 3, we can clearly understand that if we want to use the encoder and decoder followed by the attention mechanism to either describe the contents of a natural image or use this for the medical images, the combination of VGG for feature extraction and LSTM for sentence generation yielded state-of-the-art results. Although previous research efforts have used bi-directional LSTM in which we have information of the past as well as the future so that our model can predict better, it works better mostly in the caption generation of natural images [38].

**Table 3. Comparison of proposed technique with different combination of available options in terms of BLEU score up to n gram for the medical report generated on the IU CXR dataset.**

| Models | BLEU-1 | BLEU-2 | BLEU-3 | BLEU-4 |
|---|---|---|---|---|
| **VGG16 + LSTM with Attention** | **0.580** | **0.342** | **0.263** | **0.155** |
| VGG16 + LSTM without Attention | 0.522 | 0.262 | 0.201 | 0.119 |
| VGG16 + GRU | 0.495 | 0.302 | 0.250 | 0.160 |
| VGG16 + Bi-Directional LSTM | 0.533 | 0.321 | 0.253 | 0.153 |

**Table 4. Comparison of proposed technique with existing state of the art in terms of BLEU score up to n gram for the medical report generated on the IU CXR dataset.**

| Dataset | Methods | BLEU-1 | BLEU-2 | BLEU-3 | BLEU-4 |
|---|---|---|---|---|---|
| IU X-Ray | LRCN [34] | 0.369 | 0.229 | 0.149 | 0.099 |
| | Soft ATT [35] | 0.399 | 0.251 | 0.168 | 0.118 |
| | ATT-RK [36] | 0.369 | 0.226 | 0.151 | 0.108 |
| | Hierarchical Generation [37] | 0.437 | 0.323 | 0.221 | 0.172 |
| | Conditioned Transformers [39] | 0.347 | 0.221 | 0.156 | 0.116 |
| | CDGPT$^2$[39] | 0.387 | 0.245 | 0.166 | 0.111 |
| | **CNN LSTM (With Attention)** | **0.580** | **0.342** | **0.263** | **0.155** |

**Table 5. Comparison of proposed technique with current state of the art techniques in terms of BLEU-4 for the medical report generated on the MIMIC-CXR dataset.**

| | TieNet[16] | CNN-RNN$^2$[40] | R2Gen [41] | Meshed Memory Trans [42] | Proposed |
|---|---|---|---|---|---|
| BLEU-4 | 0.081 | 0.076 | 0.086 | 0.133 | **0.153** |

For the report generation, it is clear that the proposed architecture of CNN-LSTM with the unrolled LSTM format followed by the attention mechanism performs much better than all the other mentioned networks, as shown in Table 4. The contrast between these models explicitly indicates the effectiveness of the proposed CNN LSTM model. This finding is not unexpected, as it is already well established that a single-layer LSTM cannot model long sentences effectively [35]. Although, Alfarghaly et al. [39] methods resulted in relatively lower BLEU scores but they added extra information from IU X-Ray dataset related to image tags and they also assigned final automated tags to images along with report generation.

The proposed algorithm is also tested on MIMIC dataset and results are presented in terms of BLEU-4 score. Table 5 shows comparison of proposed technique with state of art methods who have used MIMIC CXR dataset. Meshed memory transform presented in [42] gives almost comparable results with proposed technique. This meshed memory model was optimized with help of 5 loss functions. This along with proposed technique clearly outperformed simple CNN and RNN based models. The attention heads introduced in proposed technique clearly helped it in getting state of the art results.

The model was trained on Google Colab, which provides a 1x NIVIDIA Tesla K80 GPU with 12 GB GDDR5 VRAM. The loss calculated for the LSTM+ VGG model was less than that for the other two models. LSTM took more time than the GRU in processing. This is due to the lesser number of operations occurring in the GRU than in the LSTM. GRUs generally train faster on less training data than LSTMs and are simpler and easier to modify.

Figs 6 and 7 represent the accuracy and loss graph of the proposed model between number of epochs, respectively. We can clearly see that after the 7th epoch, the loss starts to increase in

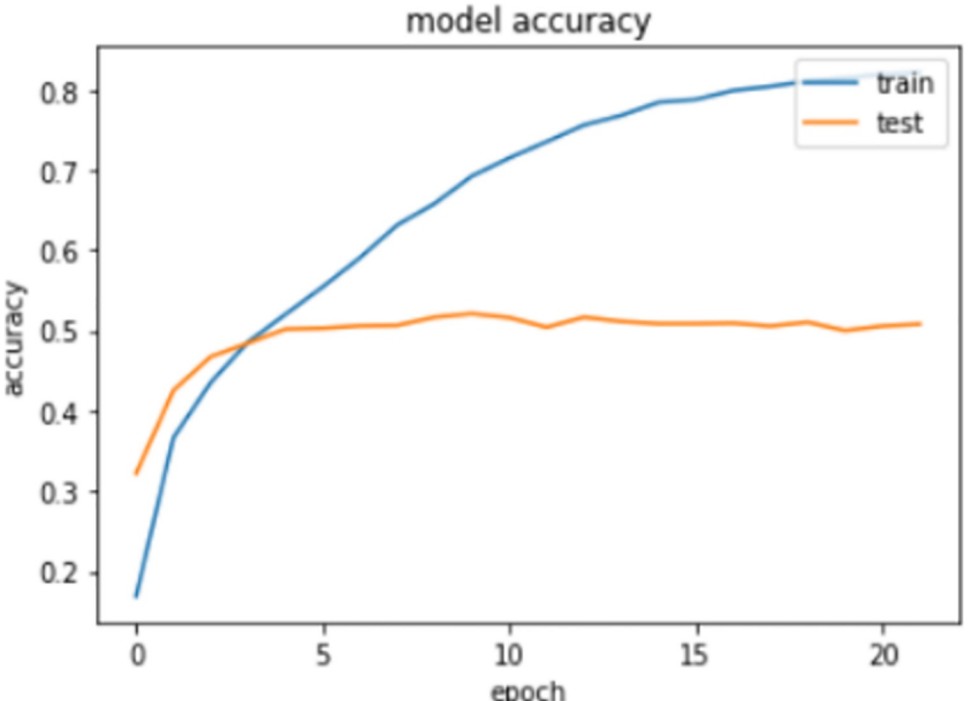

**Fig 6. Graph between accuracy and epochs using proposed model.**

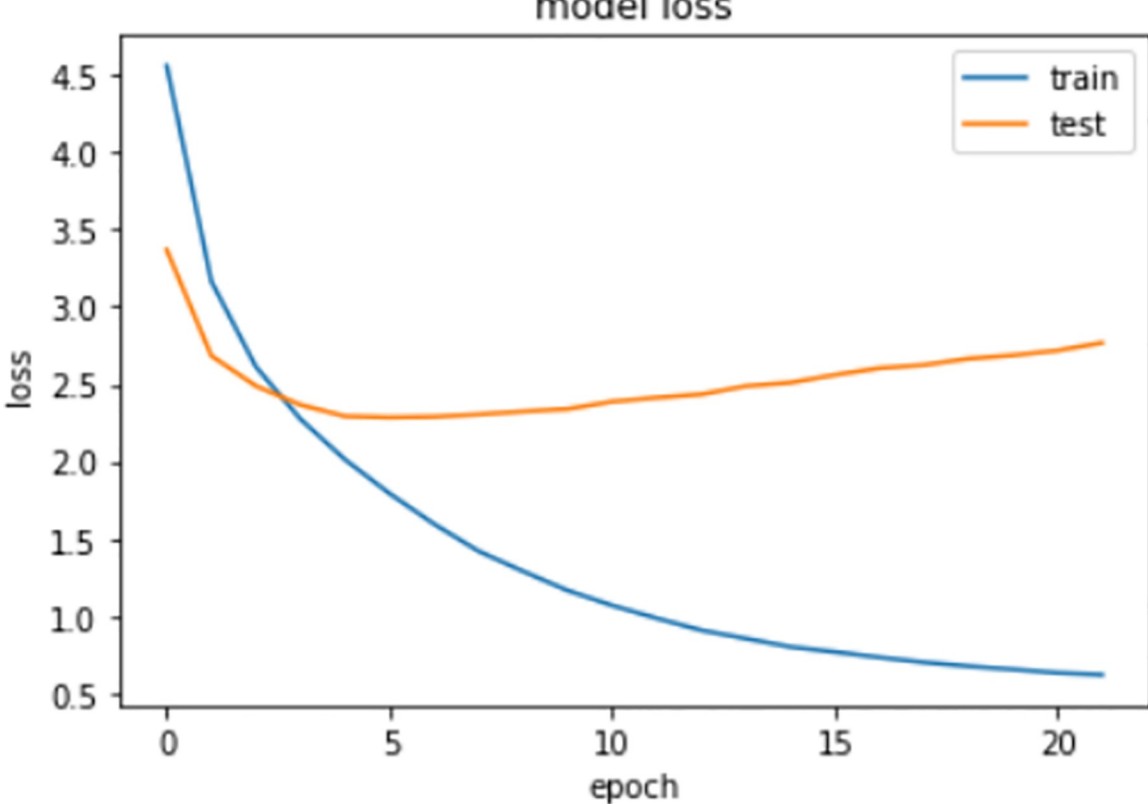

**Fig 7. Graph between loss and epochs using proposed model.**

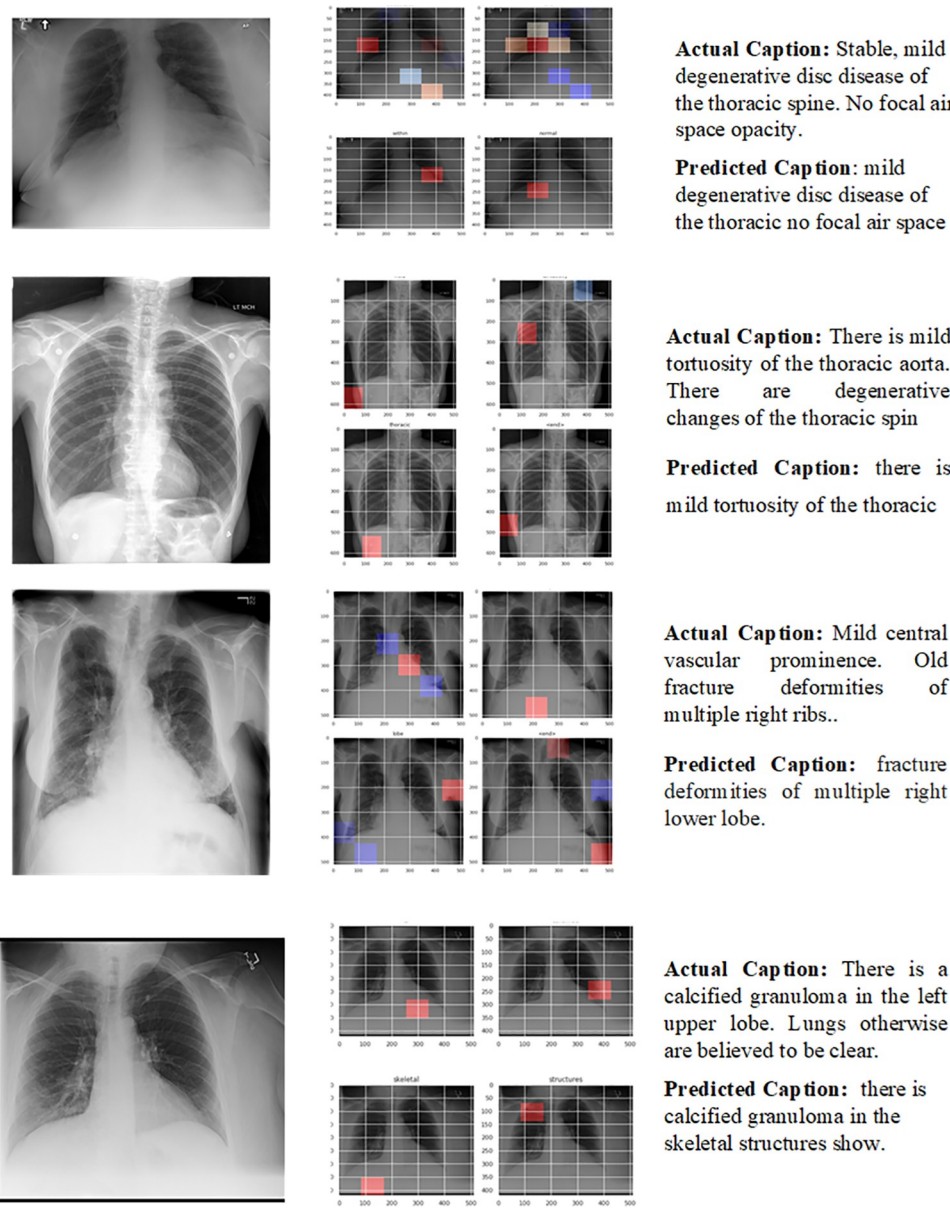

**Fig 8. Different examples of generated results.** Column 1 contains the original image, column 2 contain the attention maps and column 3 contain the actual and predicted captions.

the testing data. However, on the training data, the loss will decrease, thus showing the problem of overfitting. Therefore, to avoid this overfitting, we decided to use the trained model of the 7[th] epoch, which is the minimum loss our model can attain while training on the CXR images of the IU dataset.

The performance is also expected to increase when using a larger dataset for training on a greater number of CXRs. Because of the considerable accuracy of the generated medical reports, radiologists can gain more assistance and benefits in terms of the rapid generation of reports.

| | | BLEU-1: 0.75 | BLEU-2: 0.72 | BLEU-3: 0.60 | BLEU-4: 0.64 |
|---|---|---|---|---|---|
| | Actual | The heart size and pulmonary vascularity appear within normal limits. The lungs are free of focal airspace disease. No pleural effusion or pneumothorax is seen. | | | |
| | Predicted | The heart size and pulmonary vascularity appear within normal limits. The lungs are free of focal airspace disease. No pleural effusion or pneumothorax is seen. Calcified granuloma is identified. | | | |
| | | BLEU-1: 0.71 | BLEU-2: 0.64 | BLEU-3: 0.46 | BLEU-4: 0.49 |
| | Actual | The heart size is normal. The mediastinal contour is within normal limits. There is a streaky opacity within the right upper lobe. There are no nodules or masses. No visible pneumothorax. No visible pleural fluid. The are grossly normal. There is no visible free intraperitoneal air under the diaphragm. | | | |
| | Predicted | The heart size is normal. The mediastinal contour is within normal limits. The lungs are free of any focal infiltrates. there are no nodules or masses. No visible pneumothorax. No visible pleural fluid. The are grossly normal. There is no visible free intraperitoneal air under the diaphragm. | | | |
| | | BLEU-1: 0.02 | BLEU-2: 0.0086 | BLEU-3: 9.1e-170 | BLEU-4: 2.51e-154 |
| | Actual | On the right there is marked narrowing of the hip joint space uniformly throughout. Osteophyte formation is present with some sclerosis and subchondral cyst formation vertically along the superior acetabulum and femoral head. I do not see evidence for fracture or destructive process. AP view of the femur shows no femoral destructive process or other significant abnormality. For of the Left hip shows near-complete obliteration of the joint space with severe subchondral sclerosis and cystic formation in both the superior acetabulum and superior aspect of the femoral head. No fracture or destructive process is identified. Surgical markers were in the images and left hip for the purpose of surgical planning. PA and lateral chest show the lungs to be clear. There may be some hyperinflation. No pleural effusion is identified. The heart is normal in size. There are calcified mediastinal lymph. The skeletal structures appear normal. | | | |
| | Predicted | The trachea is midline. cardio mediastinal silhouette is normal. The lungs are clear without evidence of acute infiltrate or effusion. There is no pneumothorax. the visualized bony structures reveal no acute abnormalities. | | | |

**Fig 9. Randomly selected CXR from datasets used along with original and predicted reports.**

## 4.5 Qualitative results

Some of the sample reports generated by the proposed CNN-LSTM-based model are shown in Fig 8. The medical reports are high-level descriptions of the X-ray. Sentences generated based on different diseases present in the radiograph depend upon the features extracted by VGG or the encoder part. Many true abnormalities present in the X-ray are correctly described by the CNN-LSTM model, as shown below. Any sentence composed of the words like "no," "normal," "clear," "stable" is considered as "normality."

The performance of proposed system in terms of BLEU scores along with predicted labels is shown in Fig 8. Here we have added few randomly selected best and worst cases of CXRs from datasets used for experimentations. Here it is evident from first 2 examples of Fig 8 that proposed technique has been able to capture variations in the labels and predict the labels with good BLEU scores. However, the scores are quite low for the cases where reports are too long as shown in example 3 of Fig 9.

## 5. Conclusion

The proposed model is an application to create automated textual reports for CXR, with the aim of assisting medical professionals in creating reports more efficiently and effectively. It is based on a CNN feature extraction model that acts as an encoder that converts an image into a fixed act as an encoder that converts an image into a fixed-size vector representation, followed by an RNN decoder that generates corresponding sentences based on the learned image features. The effectiveness of the model was analyzed quantitatively and qualitatively on the CXR dataset. A comparative study of various methods has been presented to observe the influence of different components on medical report generation and has also demonstrated various use cases on the proposed system. The results show that the LSTM model generally works slightly better than GRU although it takes a little more time for the training as well as for the sentence generation owing to its complexity. The performance is also expected to increase when using a larger dataset by training on a greater number of images. Different experiments on the IU dataset validate the effectiveness of the proposed architecture.

## Author Contributions

**Conceptualization:** Mehreen Sirshar, Muhammad Faheem Khalil Paracha, Norah Saleh Alghamdi, Syeda Zainab Yousuf Zaidi.

**Data curation:** Muhammad Faheem Khalil Paracha, Tatheer Fatima.

**Formal analysis:** Muhammad Faheem Khalil Paracha.

**Investigation:** Mehreen Sirshar, Muhammad Faheem Khalil Paracha.

**Methodology:** Mehreen Sirshar, Muhammad Faheem Khalil Paracha.

**Project administration:** Muhammad Faheem Khalil Paracha.

**Resources:** Muhammad Faheem Khalil Paracha.

**Software:** Muhammad Faheem Khalil Paracha.

**Supervision:** Muhammad Usman Akram, Norah Saleh Alghamdi, Syeda Zainab Yousuf Zaidi.

**Validation:** Muhammad Faheem Khalil Paracha, Tatheer Fatima.

**Visualization:** Muhammad Faheem Khalil Paracha.

**Writing – original draft:** Muhammad Faheem Khalil Paracha.

**Writing – review & editing:** Mehreen Sirshar, Muhammad Faheem Khalil Paracha, Muhammad Usman Akram.

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
