## [Decision Letter · Decision Letter 0]

19 Aug 2021

PONE-D-21-15198

Attention Based Automated Radiology Report Generation using CNN and RNN

PLOS ONE

Dear Dr. Paracha,

Thank you for submitting your manuscript to PLOS ONE. After careful consideration, we feel that it has merit but does not fully meet PLOS ONE’s publication criteria as it currently stands. Therefore, we invite you to submit a revised version of the manuscript that addresses the points raised during the review process.

The proposed method needs to compare with other state-of-the-art methods, such as 

https://arxiv.org/abs/1912.08226

https://arxiv.org/abs/2010.10042

The experiments need to conduct on several datasets to demonstrate the model's generalizability.

The paper needs to be proofread. There are many typos and errors.

We look forward to receiving your revised manuscript.

Kind regards,

Yifan Peng, Ph.D.

Academic Editor

PLOS ONE

1. Please ensure that your manuscript meets PLOS ONE's style requirements, including those for file naming. The PLOS ONE style templates can be found at https://journals.plos.org/plosone/s/file?id=wjVg/PLOSOne_formatting_sample_main_body.pdf and https://journals.plos.org/plosone/s/file?id=ba62/PLOSOne_formatting_sample_title_authors_affiliations.pdf.

2.Thank you for stating the following in the Acknowledgments Section of your manuscript:

“This research project was funded by the Deanship of Scientific Research, Princess Nourah bint Abdulrahman University, through the “Program of Research Project Funding After Publication, grant No (42-PRFA-P-53)”.

“This research project was funded by the Deanship of Scientific Research, Princess Nourah bint Abdulrahman University, through the “Program of Research Project Funding After Publication, grant No (42-PRFA-P-53)”.

https://www.pnu.edu.sa/en/Pages/home.aspx

No, The funders had no role in study design, data collection and analysis, decision to publish, or preparation of the manuscript.”

Additional Editor Comments (if provided):

Reviewers' comments:

Reviewer's Responses to Questions

**Comments to the Author**

1. Is the manuscript technically sound, and do the data support the conclusions?

Reviewer #1: Yes

Reviewer #2: Yes

2. Has the statistical analysis been performed appropriately and rigorously? 

Reviewer #1: Yes

Reviewer #2: N/A

3. Have the authors made all data underlying the findings in their manuscript fully available?

Reviewer #1: Yes

Reviewer #2: No

4. Is the manuscript presented in an intelligible fashion and written in standard English?

Reviewer #1: Yes

Reviewer #2: No

5. Review Comments to the Author

Reviewer #1: The author proposed an attention based Automated Radiology Report Generation using CNN and RNN. The paper is easy to read and the results can confirm their idea. But there are still several issues that need to be addressed.

(1) A line is missing at the bottom of the table.

(2) The equation 2 is missing a ).

(3) The second Table 1 should be Table 2 and the Table 2 should be Table 3.

(4) The resolution of Figure 3 is low.

(5) The reference seems wrong. For example, in the section 4.3 Baselines:

The proposed model is compared with many current best performing architectures, for example, LRCN [18] by Donahue et al., Soft ATT [19] by Xu et al., ATT-RK [20] by You et al., and Hieratical Generation [21] by Krause, et al.

But in the Table 2 the references are 19-22. You have to check your references carefully.

(6) You need to include the latest method to compare with your proposed method, such as Meshed-Memory Transformer for image captioning, Linguistics Improving Factual Completeness and Consistency of Image-to-Text Radiology Report Generation. You can also compare with other methods, but at least those proposed in 2020.

Reviewer #2: This manuscript describes an attention-based report generation from X-ray using CNN and RNN.

Descriptions of x-ray images were missing from both data and experiment sections. More details on the images will be helpful (size of x-ray, divided image into n parts? what is n in the experiment?)

VGG was not elaborated on Page 12.

The experiments seem a bit confusing. For example, when LSTM and RNN were used in the experiment? Is it CNN-RNN or CNN-LSTM? In text, it is mentioned CNN-RNN, but in figure, it is CNN-LSTM.

The text of the manuscript can be reduced significantly.

Please provide all the parameters for all the algorithms.

Captions for images are not available.

Problems in table caption. The numbering of tables is wrong.

6. PLOS authors have the option to publish the peer review history of their article (what does this mean?). If published, this will include your full peer review and any attached files.

Reviewer #1: No

Reviewer #2: No

---

## [Author Response · Author response to Decision Letter 0]

24 Sep 2021

Editor’s Comments & Responses

1. The proposed method needs to compare with other state-of-the-art methods, such as 

2. https://arxiv.org/abs/1912.08226

3. https://arxiv.org/abs/2010.10042

Response:

We have considered both articles and added some more relevant ones in revised version. We have also compared the results of proposed technique with these articles. Following recent articles have been added in revised version 

• Alfarghaly, O., Khaled, R., Elkorany, A., Helal, M., & Fahmy, A. (2021). Automated radiology report generation using conditioned transformers. Informatics in Medicine Unlocked, 24, 100557.

• Liu, G., Hsu, T. M. H., McDermott, M., Boag, W., Weng, W. H., Szolovits, P., & Ghassemi, M. (2019, October). Clinically accurate chest x-ray report generation. In Machine Learning for Healthcare Conference (pp. 249-269). PMLR.

• Chen, Z., Song, Y., Chang, T. H., & Wan, X. (2020). Generating radiology reports via memory-driven transformer. arXiv preprint arXiv:2010.16056.

• Miura, Y., Zhang, Y., Tsai, E. B., Langlotz, C. P., & Jurafsky, D. (2020). Improving factual completeness and consistency of image-to-text radiology report generation. arXiv preprint arXiv:2010.10042.

2. The experiments need to conduct on several datasets to demonstrate the model's generalizability.

Response:

We have added one more publically available dataset for experimentations and results are added in revised article. We have added more results and also have highlighted failure cases to show model’s generalizability. Please refer to table-3, table-5 and figure 9. 

3. The paper needs to be proofread. There are many typos and errors.

Response:

The article was initially proof read by a third party EdiTag. We have again proofread it for possible errors. 

Reviewer's Comments & Responses

1. Is the manuscript technically sound, and do the data support the conclusions?

Reviewer #1: Yes

Reviewer #2: Yes

2. Has the statistical analysis been performed appropriately and rigorously?

Reviewer #1: Yes

Reviewer #2: N/A

3. Have the authors made all data underlying the findings in their manuscript fully available?

Reviewer #1: Yes

Reviewer #2: No

Response: 

The datasets used in this article are publically available 

4. Is the manuscript presented in an intelligible fashion and written in standard English?

Reviewer #1: Yes

Reviewer #2: No

Response: 

Manuscript is updated according to the comments of the reviewers and academic Editor and all the grammatical and typographical errors are catered. It has been proof read by third party. 

5. Review Comments to the Author

Reviewer #1

The author proposed an attention based Automated Radiology Report Generation using CNN and RNN. The paper is easy to read and the results can confirm their idea. But there are still several issues that need to be addressed.

Response: We are thankful to the reviewer for encouragement and also sharing his comments to improve the quality of our article 

(1) A line is missing at the bottom of the table.

Response: We have updated the formatting of all tables and bottom lines are added at the end of all tables. We have also updated captions of all tables for better understanding. 

(2) The equation 2 is missing a ).

Response: The equation is updated and ‘)’ is added at the end of equation 2.

(3) The second Table 1 should be Table 2 and the Table 2 should be Table 3.

Response: We have corrected table numbering.

(4) The resolution of Figure 3 is low.

Response: We have updated Figure 3 to improve its resolution. We have also improved the resolution of other figures. 

(5) The reference seems wrong. For example, in the section 4.3 Baselines:

The proposed model is compared with many current best performing architectures, for example, LRCN [18] by Donahue et al., Soft ATT [19] by Xu et al., ATT-RK [20] by You et al., and Hieratical Generation [21] by Krause, et al. But in the Table 2 the references are 19-22. You have to check your references carefully.

Response: References are updated and now all the references are properly aligned. 

(6) You need to include the latest method to compare with your proposed method, such as Meshed-Memory Transformer for image captioning, Linguistics Improving Factual Completeness and Consistency of Image-to-Text Radiology Report Generation. You can also compare with other methods, but at least those proposed in 2020.

Response: As per reviewer suggestion, we have conducted more experiments and compared our results with more recent articles. We have also considered more public datasets like MIMIC for experiments and comparisons. 

Reviewer #2:

This manuscript describes an attention-based report generation from X-ray using CNN and RNN. Descriptions of x-ray images were missing from both data and experiment sections. More details on the images will be helpful (size of x-ray, divided image into n parts? what is n in the experiment?) 

Response: We have added these details with respect to both dataset and experimental setup in section 4.2 of revised article. 

VGG was not elaborated on Page 12.

Response: Section 3.1 of revised article contains details related to VGG and we have also shown all layers of VGG in figures 2 and 3. The Softmax layer of VGG16 was replaced with the final 1 × 4096 FC layer. This layer now acts as an input to the decoder as well as the generation of medical reports. The output of the VGG16 network is a vector of size 1 × 4096, which will later be converted into a fixed vector length of 1 × 256 that is used to represent the features of the images.

The experiments seem a bit confusing. For example, when LSTM and RNN were used in the experiment? Is it CNN-RNN or CNN-LSTM? In text, it is mentioned CNN-RNN, but in figure, it is CNN-LSTM.

Response: Long Short-Term Memory (LSTM) networks are a modified version of recurrent neural networks (RNN) which makes it easier to remember past data in memory and considered best of the text generation tasks. The vanishing gradient problem of RNN is resolved here. So as a big picture we write RNN every where even in the title of the paper. But actually, we are using the best model of RNN which is LSTM. But as highlighted by the reviewer, we have replaced RNN with LSTM wherever we are mentioning proposed technique. We have also updated the title accordingly. 

The text of the manuscript can be reduced significantly.

Response: As advised by the reviewer, we have reduced the text in introduction and literature review sections. However, more results and discussions are added to emphasize more on proposed system robustness. So in general the length of article is almost same. 

Please provide all the parameters for all the algorithms.

Response: We have added parameters in figure 2,3 and 4 of revised article. We have also added table 2 to show hyper parameters. 

Captions for images are not available. 

Report: As per journal formatting, we didn’t add figures in the text. However the image captions are added at relevant places for better understanding. 

Problems in table caption. The numbering of tables is wrong.

Response: We have corrected table numbering and have also improved the captions for better understanding.

---

## [Decision Letter · Decision Letter 1]

8 Nov 2021

PONE-D-21-15198R1Attention Based Automated Radiology Report Generation using CNN and LSTMPLOS ONE

Dear Dr. Paracha,

Thank you for submitting your manuscript to PLOS ONE. After careful consideration, we feel that it has merit but does not fully meet PLOS ONE’s publication criteria as it currently stands. Therefore, we invite you to submit a revised version of the manuscript that addresses the points raised during the review process.

1. please briefly describe the baseline methods

2. please check figure 4

3. please make sure the codes are publicly available.

We look forward to receiving your revised manuscript.

Kind regards,

Yifan Peng, Ph.D.

Academic Editor

PLOS ONE

Journal Requirements:

Reviewers' comments:

Reviewer's Responses to Questions

**Comments to the Author**

1. If the authors have adequately addressed your comments raised in a previous round of review and you feel that this manuscript is now acceptable for publication, you may indicate that here to bypass the “Comments to the Author” section, enter your conflict of interest statement in the “Confidential to Editor” section, and submit your "Accept" recommendation.

Reviewer #1: All comments have been addressed

Reviewer #2: All comments have been addressed

2. Is the manuscript technically sound, and do the data support the conclusions?

Reviewer #1: Yes

Reviewer #2: Yes

3. Has the statistical analysis been performed appropriately and rigorously? 

Reviewer #1: Yes

Reviewer #2: No

4. Have the authors made all data underlying the findings in their manuscript fully available?

Reviewer #1: Yes

Reviewer #2: Yes

5. Is the manuscript presented in an intelligible fashion and written in standard English?

Reviewer #1: Yes

Reviewer #2: Yes

6. Review Comments to the Author

Reviewer #1: All my previous comments have been addressed. I have one more comments for this revised version. As the author have conducted more experiments and compared their results with more recent articles, they need added the new comparison methods to the Section Baselines.

Reviewer #2: The authors addressed all the comments.

Figure 4 is black and there is nothing.

The authors may share the code.

7. PLOS authors have the option to publish the peer review history of their article (what does this mean?). If published, this will include your full peer review and any attached files.

Reviewer #1: No

Reviewer #2: No

---

## [Author Response · Author response to Decision Letter 1]

23 Nov 2021

Comment: 

Please briefly describe the baseline methods

Response: 

We have added brief description of baseline methods in section “4.3 Baseline” of revised article. 

Comment:

Please check figure 4

Response:

Figure 4 is updated. Now it is properly visible. 

Comment:

Please make sure the codes are publicly available.

Response:

We have made our complete code publically available at our research group website. It can be accessed at http://biomisa.org/index.php/downloads/

---

## [Editor Report · Decision Letter 2]

16 Dec 2021

PONE-D-21-15198R2Attention Based Automated Radiology Report Generation using CNN and LSTMPLOS ONE

Dear Dr. Paracha,

Thank you for submitting your manuscript to PLOS ONE. After careful consideration, we feel that it has merit but does not fully meet PLOS ONE’s publication criteria as it currently stands. Therefore, we invite you to submit a revised version of the manuscript that addresses the points raised during the review process.

We look forward to receiving your revised manuscript.

Kind regards,

Yifan Peng, Ph.D.

Academic Editor

PLOS ONE

Journal Requirements:

Additional Editor Comments (if provided):

1. Figure 9 on page 39 is black

2. Figure 4 on page 40 is black

3. I feel the table on page 12 is not Fig 4.
---

## [Author Response · Author response to Decision Letter 2]

18 Dec 2021

We have made all updates with respect to figures mentioned by the editor

---

## [Editor Report · Decision Letter 3]

20 Dec 2021

Attention Based Automated Radiology Report Generation using CNN and LSTM

PONE-D-21-15198R3

Dear Dr. Paracha,

We’re pleased to inform you that your manuscript has been judged scientifically suitable for publication and will be formally accepted for publication once it meets all outstanding technical requirements.

Kind regards,

Yifan Peng, Ph.D.

Academic Editor

PLOS ONE
---

## [Editor Report · Acceptance letter]

27 Dec 2021

PONE-D-21-15198R3 

Attention based Automated Radiology Report Generation using CNN and LSTM 

Dear Dr. Paracha:

I'm pleased to inform you that your manuscript has been deemed suitable for publication in PLOS ONE. Congratulations! Your manuscript is now with our production department. 

Kind regards, 

on behalf of

Dr. Yifan Peng 

Academic Editor

PLOS ONE